# Bleeding Patterns of Oral Contraceptives with a Cyclic Dosing Regimen: An Overview

**DOI:** 10.3390/jcm11154634

**Published:** 2022-08-08

**Authors:** David F. Archer, Diana Mansour, Jean-Michel Foidart

**Affiliations:** 1Department of Obstetrics and Gynecology, Clinical Research Center, Eastern Virginia Medical School, Norfolk, VA 23507, USA; 2Department of Sexual Health, New Croft Centre, Newcastle Hospitals, Community Health, Newcastle upon Tyne NE1 6ND, UK; 3Estetra SRL, Mithra Pharmaceuticals, 4000 Liège, Belgium; 4Department of Obstetrics and Gynecology, University of Liège, 4000 Liège, Belgium

**Keywords:** oral contraceptives, bleeding pattern, ethinylestradiol, estetrol, estradiol, drospirenone, norethindrone acetate, nomegestrol acetate, dienogest

## Abstract

Bleeding irregularities are one of the major reasons for discontinuation of oral contraceptives (OCs), and therefore clinicians need to set expectations during consultations. In this review we provide an overview of bleeding data of recently marketed cyclic combined OCs (COCs) and one progestin-only pill (POP). We evaluated data from phase 3 trials (≥12 months) used to gain regulatory approval. Overall, each type of OC has its own specific bleeding pattern. These patterns however were assessed by using different bleeding definitions, which hampers comparisons between products. In COCs, the estrogen balances the effects of the progestin on the endometrium, resulting in a regular bleeding pattern. However, this balance seems lost if a too low dose of ethinylestradiol (EE) (e.g., 10 µg in EE/norethindrone acetate 1 mg) is used in an attempt to lower the risk of venous thromboembolism. Replacement of EE by 17β-estradiol (E2) or E2 valerate could lead to suboptimal bleeding profile due to destabilization of the endometrium. Replacement of EE with estetrol (E4) 15 mg in the combination with drospirenone (DRSP) 3 mg is associated with a predictable and regular scheduled bleeding profile, while the POP containing DRSP 4 mg in a 24/4 regimen is associated with a higher rate of unscheduled and absence of scheduled bleeding than combined products.

## 1. Introduction

Bleeding irregularities, including absence of scheduled bleeding (withdrawal bleeding), changes in volume of blood loss or duration of scheduled bleeding, and bleeding (including spotting) on unexpected days (unscheduled bleeding), are a well-known side effect of hormonal contraception. Such bleeding irregularities are often reported at the start of contraceptive use, especially by first time users, and diminish or disappear after the first three months [1,2]. However, it is one of the major reasons to discontinue oral contraceptives [1,2,3]. The occurrence of bleeding irregularities depends on the type of oral contraceptive (OC) [4] but may also result from a lack of treatment adherence, interactions with other drugs or intercurrent illness, such as vomiting and diarrhea. Moreover, smoking and a high body mass index (BMI) increase the likelihood of bleeding irregularities [5,6].

Hormonal contraceptives contain either estrogen plus progestin (combined oral contraceptives, COCs) or progestin only (progestin-only pills, POPs). While progestins are primarily responsible for the contraceptive effect, the estrogenic component of a COC balances the effects of the progestin on the endometrium, resulting in a regular bleeding pattern. Currently, the majority of COCs contain the synthetic estrogen ethinylestradiol (EE). Over the years, EE has been reduced to doses as low as 10 µg in order to lower the risk of venous thromboembolism (VTE). Novel COCs have been introduced, replacing EE with 17β-estradiol (E2), its esterified prodrug E2 valerate (E2V), or by the fetal estrogen estetrol (E4), in an attempt to reduce the cardiovascular risk. For users of E2-containing COCs, the VTE risk appears slightly lower compared to levonorgestrel/EE-containing products [7,8]. For the E4-containing COC, real life data on VTE risk are not yet available, but the effects on hemostasis parameters are minimal [9].

Changes in the estrogen component of a COC, may change the stabilizing estrogenic effect on the endometrium and may consequently affect the bleeding pattern. In 2007, Bachmann and Korner [10] reviewed the bleeding patterns of OCs in clinical trials published between 1986 and 2006. They highlighted the need to use standardized reporting of bleeding patterns. Up to then, different methods were used in clinical research to analyze the bleeding patterns associated with hormonal contraceptive use. In 1986, the World Health Organization (WHO) published a standard with recommendations on collection and analysis of bleeding data based on a 90-day reference period, also referred to as the Belsey criteria [11]. The 90-day reference period defined in this standard does not distinguish between scheduled and unscheduled bleeding, which is one of the main reasons why these criteria were not uniformly applied [12]. In 2007, a new set of recommendations for standardization of data collection, study design, and analysis of bleeding data was published by Mishell et al. [13], with new definitions for bleeding parameters, including definitions for unscheduled and scheduled bleeding (Table 1).

Since the publication of Mishell et al. [13], new OCs have entered the market. In this review, we provide an overview of the bleeding data for recently marketed cyclic OCs.

## 2. Methods

We present an overview of bleeding data from phase 3 clinical trials with four cyclic COCs: estradiol valerate 1–3 mg/dienogest 2–3 mg (E2V/DNG Bayer Healthcare Pharmaceuticals, Wayne, NJ, USA), EE 10 µg/norethindrone acetate 1 mg (EE/NETA Abbvie, Chicago IL, USA), estradiol 1.5 mg/nomegestrol acetate 2.5 mg (E2/NOMAC Theramex Paris, France), and estetrol 15 mg /drospirenone 3 mg (E4/DRSP; Estetra SRL, Liège, Belgium), and of one cyclic POP: 4 mg DRSP-only (Exeltis Healthcare Madrid, Spain). The phase 3 trials with E2/NOMAC included the internal comparator drospirenone 3 mg/EE 30 µg (EE/DRSP), in a 21/7 treatment regimen (Table 2 and Table 3). Table 2 and Table 3 and Figure 1 also include data from a phase 3 trial with DRSP/EE in a 24/4 treatment regimen, as a historical reference for a 24/4 regimen, and the data of a phase 2 trial with E4/DRSP that included E2V/DNG as an internal comparator.

For the overview, we collected bleeding data from phase 3 contraceptive trials (with a duration of at least 12 months) that were used to gain regulatory approval. From public assessment reports of regulatory agencies, we gathered data on the incidence of scheduled and unscheduled bleeding per treatment cycle and on the number of bleeding days by reference period. Public assessment reports may contain more complete and correct data, as publications can come out before approval of the product in which final corrections are not yet incorporated. For this reason, the use of these data can add to the quality of reviews [14]. Based on a search of the FDA/EMA and the Australian regulatory Therapeutic Goods Administration database, we could retrieve the assessment reports of all five cyclic oral contraceptives. In addition to the reports from regulatory agencies, we used data from scientific publications on the individual or pooled phase 3 clinical trials. For this, we performed targeted searches in PubMed to find any peer-reviewed clinical articles published between July 2006 and April 2022 that were related to the phase 3 contraceptives trials used to gain regulatory approval. The following search terms were used: oral contraception, oral contraceptive, bleeding, bleeding pattern, phase 3, drospirenone, nomegestrol, dienogest, norethindrone acetate, ethinylestradiol, estradiol, estradiol valerate, and estetrol. As search terms such as “phase 3” and “bleeding” were not always used as keywords for the phase 3 clinical trials that were used for registration, we did not perform a systematic search, but emphasis was placed on articles that could add useful clinical information regarding the bleeding profile in these phase 3 contraceptive trials. In addition, any literature references mentioned in clinicaltrials.gov register [15] related to these phase 3 clinical trials were assessed. We found 13 publications on the phase 3 clinical trials that provided detailed information on the bleeding profile and included those in this overview. Table 2 provides an overview of the trials and sources (public assessment reports and scientific publications) used in this review.

## 3. Results—Overview of Bleeding Patterns of Most Recently Marketed Cyclic OCs

All phase 3 clinical trials described here, except those with E4/DRSP and DRSP-only, started before publication of the Mishell definitions [13]. This implies that different methods were used for the collection methods, the definition of bleeding, and the type of analyses performed across the slected clinical trials. For all products, an expected bleeding period is defined, and bleeding outside this period is considered to be unscheduled bleeding, but the exact definitions of the characteristics of these periods are quite different (Table 3). Furthermore, the definitions used for bleeding and spotting were also different. In some trials, the use of panty liners was considered as bleeding (E4/DRSP and DRSP-only) while in the other trials the use of panty liners was considered as spotting (Table 3).

The term bleeding was often used for the combination of all bleeding events, including spotting, considering that spotting is a mild form of bleeding. Overall, this observed difference in methods for the assessment of bleeding across the different oral contraceptives means that comparisons between products should not be made, with the exception of when two products are used in the same trial.

Figure 1 presents the incidence of scheduled bleeding, unscheduled bleeding, and absence of scheduled bleeding by cycle for each of the different OCs displayed above each subfigure. Although unscheduled bleeding data for Cycle 1 are included in the figure, it should be noted that this cycle is not evaluable for unscheduled bleeding, as in some instances, unscheduled bleeding is expected, especially in starters who initiate treatment on their first day of menstruation.

**Figure 1 jcm-11-04634-f001:**
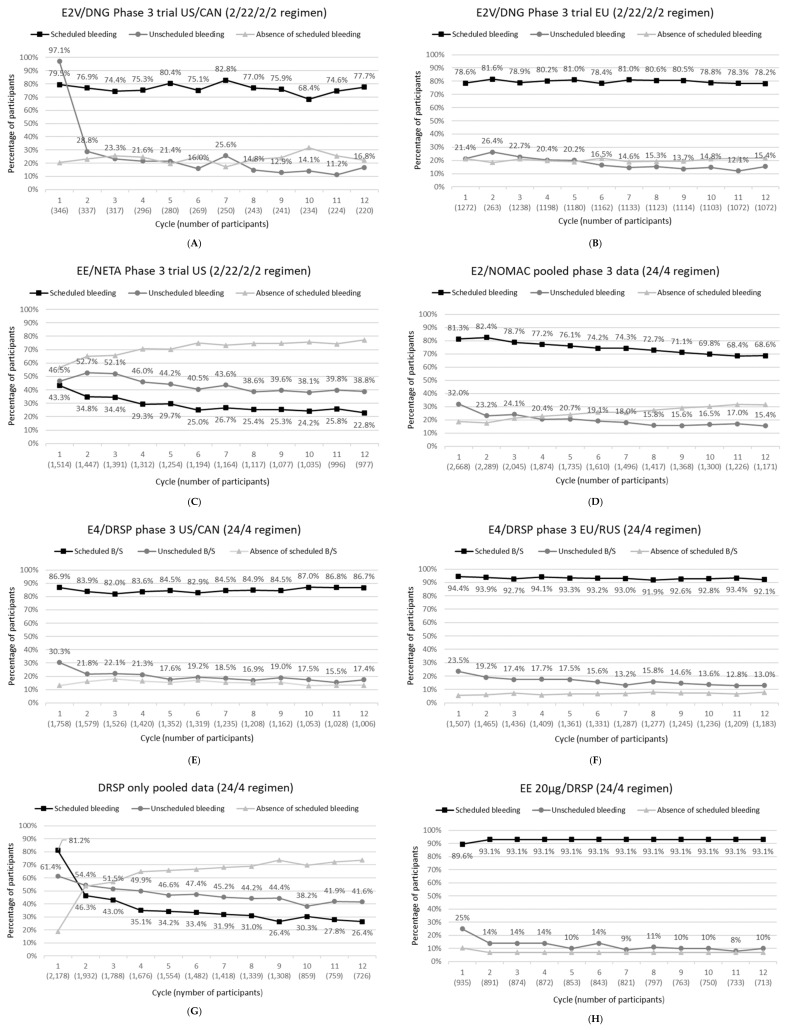
(**A**–**H**): Incidence of scheduled bleeding, unscheduled bleeding and absence of scheduled bleeding for each OC listed above each subfigure as a function of cycles.

Footnote for Figure 1: Data included: E2V/DNG trials NCT00206583 and NCT00185289; EE/NETA trial NCT00391807; E2/NOMAC pooled data of trials NCT00511199l and NCT00511199; E4/DRSP trials NCT02817828 and NCT02817841; DRSP only: pooled data trials EudraCT 2010-021787-15, EudraCT 2011-002396-42, NCT02269241 and EudraCT 2013-002300-13; EE 20 µg/DRSP trial 303740.

Meaning of abbreviations: DRSP, drospirenone; DNG, dienogest; EE, ethinylestradiol; E2: 17β-estradiol; E2V, E2 valerate; E4, estetrol; FDA, Food and Drug Administration; NETA, norethindrone acetate; NOMAC, nomegestrol acetate.

### 3.1. E2V/DNG

E2V/DNG is the first combined oral contraceptive containing esterified estradiol, which is globally registered (first approval 2008). In contrast to other OCs discussed in this manuscript, E2V/DNG is a multiphasic pill containing varying dosages of the progestin and estrogen component. Treatment cycles start with 2 days of E2V 3 mg, followed by 5 days of DNG 2 mg/E2V 2 mg, 17 days of DNG 3 mg/E2V 2 mg, 2 days of E2V 1 mg and 2 days of inactive treatment, referred to as a 2/22/2/2-day treatment regimen.

During use of E2V/DNG, scheduled bleeding was stable throughout the trial period and occurred on average over cycles 1 to 12 in 76.5% of participants in the North American trial [16] and in 79.7% of participants in the European trial [17] (absence of scheduled bleeding 23.5% and 20.3%, respectively) (Figure 1A,B). The incidence of unscheduled bleeding diminished from 28.8% in Cycle 2 to 11.2% in Cycle 11 for the North American trial [16] and from 26.4% in Cycle 2 to 12.1% in Cycle 11 in the European trial [17] (Figure 1A,B). Discontinuation rates due to bleeding-related adverse events were 2.5% in the EU trial [18] and 5.1% in the US trial [16].

In addition to the two phase 3 trials, information on the bleeding profile of E2V/DNG is provided in a 7-month comparative trial with a COC containing levonorgestrel 100 µg and EE 20 µg (LNG/EE) in a 21/7 regimen [36]. The results of this trial showed that E2V/DNG has a lower rate of scheduled bleeding (77.7% to 83.2%) compared to LNG/EE (89.5% to 93.8%) (*p* < 0.001 per cycle), which may be related to the difference in treatment regimen. The incidence of unscheduled bleeding was similar for E2V/DNG (ranging between 10.5% and 18.6%) and LNG/EE (ranging between 9.9% and 17.1%).

### 3.2. EE/NETA

EE/NETA is a COC that combines NETA 1 mg with an ultra-low dose (10 µg) of the synthetic estrogen EE in a 24/2/2-day treatment regimen. The treatment cycle of EE/NETA starts with 24 days of NETA 1 mg/EE 10 µg, followed by 2 days of EE 10 µg and 2 days of ferrous fumarate 75 mg. EE/NETA is registered in the US (approval 2010).

Use of EE/NETA resulted in a low incidence of scheduled bleeding, which decreased from 43.3% in Cycle 1 to 22.8% in Cycle 12 (Figure 1C). Absence of scheduled bleeding was reported more frequently by switchers compared to new users and by older (≥36 years) compared to younger participants [20]. Unscheduled bleeding occurred in 52.7% of participants in Cycle 2 and decreased to 38.8% in Cycle 12 (Figure 1C). Unscheduled bleeding tended to be higher in new users compared to switchers [20]. The percentage of participants discontinuing treatment due to bleeding-related adverse events was 3.8% [21].

### 3.3. E2/NOMAC

E2/NOMAC is a COC with a 24/4-day treatment regimen, with 24 tablets containing NOMAC 2.5 mg combined with E2 1.5 mg followed by 4 inactive tablets. E2/NOMAC is registered in Europe (approval 2011).

In the pooled phase 3 analysis, scheduled bleeding was reported by 82.4% of participants in Cycle 2, decreasing to 68.6% in Cycle 12 (Figure 1D), which was significantly lower compared to the reference COC EE/DRSP (ranging between 96.6% and 94.2%), an effect that may be related to the difference in treatment regimens (24/4 vs. 21/7) [23]. Absence of scheduled bleeding with E2/NOMAC was more common in participants aged ≥ 35 years (odds ratio [OR] 1.36, participants with a BMI ≥ 25 kg/m^2^ (OR 1.47), switchers (OR 1.43), and smokers (OR 1.25 [25]. Unscheduled bleeding incidence ranged from 15.4 to 24.1% over cycles 2 to 12 [23,25] (Figure 1D) and was significantly higher with E2/NOMAC compared to EE/DRSP [25]. Unscheduled bleeding appeared to be more common in starters (OR 1.38) and discontinuers (OR 1.19), but showed little association with age, BMI, and smoking [25]. During the phase 3 trials, 3.9% of participants discontinued E2/NOMAC use due to bleeding-related adverse events [25].

### 3.4. E4/DRSP

E4/DRSP is a COC containing E4, an estrogen produced in the human fetal liver [37] and manufactured for clinical use from a plant source [38,39,40]. E4/DRSP has a 24/4-day treatment regimen, with 24 tablets containing DRSP 3 mg combined with E4 15 mg (as monohydrate) followed by 4 inactive pills. E4/DRSP is globally registered (first approval 2021) [39,40].

During use of E4/DRSP in the phase 3 trials, the occurrence of scheduled bleeding was stable throughout the 12-month treatment period, with a regular scheduled bleeding in more than 92% of participants in the EU/RUS trial [26] and more than 83% of participants in the US/CAN trial [28] (Figure 1E,F). The percentage of women that reported unscheduled bleeding/spotting during treatment decreased from 19.2% in Cycle 2 to 13–18% from Cycle 3 onwards in the EU/RUS trial and from 21.8% in Cycle 2 to 15–20% from Cycle 5 onwards in the US/CAN trial (Figure 1E,F). The majority of unscheduled bleeding events were qualified by participants as spotting (not needing sanitary protection). The percentage of women that discontinued treatment due to bleeding problems was 3.4% in the EU/RUS trial and 2.7% in the US/CAN trial [26,28].

In addition to the phase 3 trials, a 6-cycle dose-finding phase 2 trial provided information on the bleeding pattern of E4/DRSP in comparison to E2V/DNG, which also contains a natural estrogen [19]. Results of this trial showed that the frequencies of unscheduled bleeding and absence of bleeding at Cycle 6 were lower with E4/DRSP (33.8% and 3.5%, respectively) compared to E2V/DNG (47.8% and 27.1%, respectively) at the end of treatment, suggesting a favorable bleeding pattern of E4/DRSP. Overall, E4/DRSP use resulted in a predictable bleeding pattern with limited unscheduled bleeding episodes, mostly defined as spotting.

### 3.5. DRSP-Only

In contrast to other available POPs that are provided as a continuous treatment, DRSP-only has a 24/4-day treatment regimen, consisting of 24 days of treatment with DRSP 4 mg, followed by 4 days of treatment with inactive tablets. The 24/4-day regimen DRSP-only was designed to reduce unscheduled bleeding, as this is a major reason for discontinuing the use of POPs. DRSP-only is globally registered (first approval 2019).

In the pooled phase 3 bleeding analysis, scheduled bleeding decreased from 81.2% in Cycle 1 and 46.3% in Cycle 2 to 26.4% in Cycle 12, while unscheduled bleeding decreased from 54.4% in Cycle 2 to 41.6% in Cycle 12 (Figure 1G). The number of participants discontinued due to bleeding irregularities was not provided for the pooled analysis but was reported to be 4.2% in the European trial [29].

The bleeding profile of the cyclic DRSP-only was compared with a typical continuous use of DSG-containing POP [31]. In this 9-cycle comparative phase 3 trial DRSP-only showed less bleeding compared to DSG. The rate of discontinuation due to bleeding-related adverse events was lower with DRSP-only (3.3%) compared to DSG (6.6%).

## 4. Discussion

In this review, we presented an overview of bleeding patterns of four cyclic COCs and the only cyclic POP marketed since 2008. We used data from public assessment reports of regulatory agencies, supplemented with data from scientific publications. In our literature search, we found that key words for bleeding, bleeding pattern, and phase 3 clinical trials were not uniformly applied, making it difficult to perform a systematic review. Instead, we chose to perform targeted searches for literature on bleeding reported during the phase 3 clinical trials of the five oral contraceptive products discussed in this review.

Overall, the more recent OCs each have their own specific bleeding pattern. It should be emphasized that based on presented data, a comparison of the different OCs cannot be made due to different definitions used in the trials. We therefore emphasize the importance to use uniform definitions for bleeding, spotting, and scheduled and unscheduled bleeding. For a detailed description of cycle control, separate reporting of bleeding and spotting episodes is recommended along with the consistent calculation of the bleeding duration, excluding users without any bleeding episodes. New recommendations on the standardization of bleeding data analysis has been published [41].

Three out of the four COCs contain a natural estrogenic component; E2V (E2V/DNG), E2 (E2/NOMAC), or E4 (E4/DRSP); the fourth COC contains an ultra-low dose of synthetic EE (EE/NETA 10 µg). The ultra-low dose of EE in EE/NETA is geared toward a safer product in an attempt to lower the risk of VTE. However, lowering the dose of EE (below 20 µg) leads to a reduced estrogenic effect on the endometrium, and a less favorable bleeding profile [42]. This is clearly reflected by the relatively high incidence of unscheduled bleeding and absence of scheduled bleeding in phase 3 trials with EE/NETA. Replacement of EE by E2 results in a suboptimal bleeding pattern due to destabilization of the endometrium. Users of E2/NOMAC experience unscheduled bleeding and absence of scheduled bleeding, although to a lesser extent compared to EE/NETA. In phase 3 trials, the use of E2V/DNG resulted in a more stable bleeding pattern, with a relatively high incidence of scheduled bleeding. This high incidence is probably ameliorated by the chosen definition of scheduled bleeding. For E2V/DNG, a scheduled bleeding episode was indeed defined as the first bleeding that started not earlier than Day 21 (i.e., not more than 4 days before DNG withdrawal) and continued without interruption. No end day of the bleeding period was defined. If no bleeding occurred until Day 20 of the next cycle, it was assessed as absence of scheduled bleeding in the previous treatment cycle. Otherwise, any bleeding occurring during the first 20 days of the next cycle caused the previous cycle to be considered as a cycle with scheduled bleeding. The rather unique definition of scheduled bleeding in studies with E2V/DNG evidently increases the proportion of scheduled bleeding episodes and makes a comparison with other OCs using Mishell’s criteria impossible. In phase 3 trials, the use of E4/DRSP resulted in a predictable bleeding profile with a regular scheduled bleeding. The cyclic DRSP-only POP lacks estrogenic activity, and in phase 3 trials, using similar evaluation criteria as E4/DRSP, this resulted in a relatively high rate of unscheduled bleeding and absence of scheduled bleeding. In general, however, comparisons can be hampered by the use of different bleeding definitions, making comparisons between products impossible.

Older COCs have a 21/7-day treatment regimen, with doses of EE 30 µg, generally resulting in a regular bleeding profile with a consistent scheduled bleeding during the 7 hormone-free days, as is observed with EE 30µg/DRSP (scheduled bleeding > 90%). The more recent products have a 24/4-day regimen, with only 4 hormone-free days. This affects the bleeding profile, especially with respect to the occurrence of scheduled bleeding. Products containing a progestin with a longer half-life, such as DRSP (t_1/2_ ~ 30 h [43]) and NOMAC (t_1/2_ ~ 46 h [44]), may result in a delay of scheduled bleeding until the first day(s) of active tablet intake of the next cycle, because the progestin levels remain high during the first hormone-free days. It is important to note that from the user’s perspective, the timing of the scheduled bleeding relative to tablet intake may not be an issue, as long as it is predictable and of acceptable length, but users should be informed about this possible delayed start of the withdrawal bleeding coinciding with the start of the next treatment cycle.

With EE/NETA and E2/NOMAC, the incidence of scheduled bleeding decreases over time, and at Cycle 12, only ~20% of participants using EE/NETA and ~70% of participants using E2/NOMAC experienced a scheduled bleeding. For E2V/DNG and E4/DRSP, the frequency of scheduled bleeding is consistent over time and appears lower with E2V/DNG (~80%) compared to E4/DRSP (~90%). While a withdrawal bleeding during OC use does not represent physiological menstruation or serve a medical need, a regular monthly bleeding may be desirable, as it reassures the user that she is not pregnant [45,46]. Women considering the use of an OC with a low frequency of scheduled bleeding should be counseled appropriately.

Nowadays, many types of contraception are available, which makes counselling of potential users on the possibilities, the risks, and the potential benefits an important factor in the prescription of contraceptives. Because bleeding irregularities are one of the major reasons for discontinuation, potential users should receive adequate information on the expected bleeding pattern and the risk of bleeding irregularities. The newer OCs presented here each have their own particular bleeding pattern. Data in this review may provide prescribers with useful information on the bleeding pattern of these OCs needed for counselling of potential users.

## Figures and Tables

**Table 1 jcm-11-04634-t001:** Terminology and definitions according to Mishell et al. [13].

Bleeding	Evidence of blood loss that requires the use of sanitary protection with a tampon, pad, or pantyliner.
Spotting	Evidence of minimal blood loss that does not require new use of any type of sanitary protection, including pantyliners.
Episode of bleeding	Bleeding and/or spotting days bounded on either end by 2 days of no bleeding or spotting.
Scheduled bleeding and withdrawal bleeding *	Any bleeding or spotting that occurs during hormone-free intervals regardless of the duration of the regimen and may continue into the first 4 days (Days 1–4) of the subsequent cycle of combined hormonal contraceptive (CHC) therapy.
Unscheduled bleeding ^#^	Any bleeding that occurs while taking active hormones, regardless of the duration of regimen. Two exceptions: Bleeding that begins during a hormone-free interval and continues through Days 1–4 of the subsequent active cycle should not be considered unscheduled. Bleeding reported during Days 1–7 of the first cycle of any study medication should not be considered unscheduled.
Unscheduled spotting ^#^	Any spotting that occurs while taking active hormones, regardless of the duration of regimen. Two exceptions: Spotting that begins during a hormone-free interval and continues through Days 1–4 of the subsequent active cycle should not be considered unscheduled. Spotting reported during Days 1–7 of the first cycle of any study medication should not be considered unscheduled.
Amenorrhea	Use of the term amenorrhea should be abandoned in the context of CHC therapy and replaced with absence of all bleeding and spotting.

* The use of traditional terminology (periods or menses) should be abandoned with regard to CHCs and replaced by the use of scheduled bleeding or withdrawal bleeding. Scheduled bleeding emphasizes to the woman that her bleeding with hormonal methods is not the same as menstruation. ^#^ Unscheduled means bleeding or spotting is not expected. The terms breakthrough bleeding and breakthrough spotting should be abandoned and replaced by unscheduled bleeding and unscheduled spotting.

**Table 2 jcm-11-04634-t002:** Clinical trials and pooled analyses used for presentation of the bleeding profiles.

Product/Brand Name	Treatment Regimen	Trial Identifier	Trial/ Countries	Trial Start- Completion	No. Subjects/ Age (Years)/ BMI (kg/m^2^) *	Data Sources	Discontinuation Rate (%) Overall/Due to AEs/ Due to Bleeding Related AEs
E2V/DNG Qlaira ^®^/Natazia^®^	2/22/2/2	NCT00206583	Phase 3 US/CAN	March 2005–July 2007	490/18–35/≤30	Nelson 2014 [16] FDA assessment report [17]	35.1/13.9/5.1
		NCT00185289	Phase 3 EU	April 2004–July 2006	1377/18–50/≤30	Palacios 2010 [18] FDA assessment report [17]	21.4/10.2/2.5
		NCT01221831	Phase 2 Finland (6 cycles)	September 2010–September 2011	78/18–35/18–30	Apter 2016 [19]	10.3/NR/NR
EE/NETA 10 µg Lo Loestrin ^®^	24/2/2	NCT00391807	Phase 3 US	November 2006–September 2008	1660/18–45/≤35	Archer 2013 [20] FDA assessment report [21]	41.7/10.7/3.8
E2/NOMAC Zoely^®^	24/4	NCT00511199	Phase 3 EU/Asia/AUS	May 2006–April 2008	1591/18–50/17–35	Mansour 2011 [22] AUSPAR [23]	28.2/18.2/4.0
		NCT00413062	Phase 3 US/LA/CAN	June 2006–July 2008	1666/18–50/17–35	Westhoff 2012 [24] AUSPAR [23]	40.7/17.3/3.8
		NCT00511199 NCT00413062	Pooled analysis	-	3233/18–50/17–35	Mansour 2017 [25]	34.7/17.9/3.9
E4/DRSP Nextstellis^®^/ Drovelis^®^/ Lydisilka^®^	24/4	NCT02817828	Phase 3 EU/RUS	June 2016–April 2018	1553/18–50/≤35	Gemzell Danielsson 2022 [26] FDA assessment report [27]	21.6//10.1/3.4
		NCT02817841	Phase 3 US/CAN	August 2016–November 2018	1864/16–50/≤35	Creinin 2021 [28] FDA assessment report [27]	45.5/9.7/2.7
		NCT01221831	Phase 2 Finland (6 cycles)	September 2010–September 2011	79/18–35/18–30	Apter 2016 [19]	8.9/NR/NR
DRSP-only Slynd^®^/Slinda^®^	24/4	2010-021787-15	Phase 3 Europe	July 2011–March 2013	713/18–45/-	Archer 2015 [29] FDA assessment report [30]	27.8/12.3/4.2
		2011-002396-42	Phase 3 Europe (9 cycles)	August 2012–January 2014	858/18–45/-	Palacios 2020 [31] FDA assessment report [30]	19.8/9.6/3.3
		NCT02269241	Phase 3 US	October 2014–October 2017	1006/≥15/-	FDA assessment report [30] Kimble 2020 [32]	65.0/11.2/NR
		EudraCT 2010-021787-15 EudraCT 2011-002396-42 NCT02269241 EudraCT 2013-002300-13	Pooled analysis	-	2598/≥15/-	FDA assessment report [30]	39.5/10.9/NR
EE 30 µg/DRSP Yasmin^®^	21/7	NCT00511199	Phase 3 EU/Asia/AUS	May 2006–April 2008	535/18–50/17–35	Mansour 2011 [22] AUSPAR [33]	23.4/10.5/0.7
		NCT00413062	Phase 3 US/LA/CAN	June 2006–July 2008	554/18–50/17–35	Westhoff 2012 [24] AUSPAR [33]	37.9/10.1/1.8
		NCT00511199 NCT00413062	Pooled analysis	-	1084/18–50/17–35	Mansour 2017 [25]	30.9/10.3/1.3
EE 20 µg/DRSP Yaz^®^	24/4	Protocol 303740	Phase 3 US/EU/LA		1027/17–36/≤35	FDA assessment report [34] Bachman 2004 [35]	28.9/7.5/NR

AE, adverse event; AUSPAR; Australian Public Assessment Report for prescription medicines; BMI, body mass index; DRSP, drospirenone; DNG, dienogest; EE, ethinylestradiol; E2: 17β-estradiol; E2V, E2 valerate; E4, estetrol; FDA, Food and Drug Administration; NETA, norethindrone acetate; No, number; NOMAC, nomegestrol acetate; NR, not reported. * Criteria on age and BMI as provided by the investigators.

**Table 3 jcm-11-04634-t003:** Heterogeneity of bleeding/spotting data collection and definition in the phase 3 trials of the different OCs evaluated.

Product/Brand Name	Trial Identifier(s)	Data Collection	Definition of Spotting	Definition of Bleeding	Definition of Scheduled Bleeding
E2V/DNG Qlaira^®^/Natazia^®^	NCT00206583 NCT00185289	Paper diary	Vaginal blood in an amount that did not require the use of sanitary protection other than panty liner(s)	Vaginal bleeding the subject categorized as light, normal, or heavy, that, based on her personal experience, required the use of sanitary protection	Start at day 21 and continued without interruption. If no bleeding occurred until Cycle Day 20 of the next cycle, it was assessed as absence of scheduled bleeding in the previous treatment cycle.
EE/NETA 10 µg Lo Loestrin^®^	NCT00391807		Light bleeding not requiring sanitary protection (other than panty liners) was classified as spotting	Only spotting defined. All vaginal bleeding was recorded, including use of sanitary protection (other than panty liners)	First bleeding episode starting 4 days before the last day of active drug intake during a treatment cycle and 3 days after beginning treatment in the next treatment cycle.
E2/NOMAC Zoely^®^	NCT00511199 NCT00413062	Electronic diary	Vaginal bleeding requiring none or at most one pad/tampon per day	Vaginal bleeding requiring more than one pad/tampon per day	Within expected bleeding period (Day 25-Day 3 next cycle) including early and continued bleeding *.
E4/DRSP Nextstellis^®^/ Drovelis^®^/ Lydisilka^®^	NCT02817828 NCT02817841	Paper diary	Minimal vaginal blood loss that did not require the new use of sanitary protection, including panty liners	Vaginal blood loss that required the use of sanitary protection with a tampon, pad or panty liner	Within expected bleeding period (Day 25-Day 3 next cycle) including early and continued bleeding *.
DRSP only Slynd^®^/Slinda^®^	EudraCT 2010-021787-15 EudraCT 2011-002396-42 NCT02269241	Electronic diary	Blood loss that did not require new use of any type of sanitary protection	Blood loss that required the use of sanitary protection	Any bleeding or spotting that occurred during hormone-free intervals (defined as days 25–28 ± 1). Up to eight consecutive bleeding days were considered as scheduled bleeding days.
EE 30 µg/DRSP Yasmin^®^	NCT00511199 NCT00413062	Electronic diary	Vaginal bleeding requiring none or at most one pad/tampon per day	Vaginal bleeding requiring more than one pad/tampon per day	Within expected bleeding period (Day 21-Day 28 next cycle) including early and continued bleeding *.
EE 20 µg/DRSP Yaz^®^ 24/4 regimen	Protocol 303740	Paper diary	No requirement for sanitary protection except panty liners	Need for sanitary protection	First bleeding after hormone withdrawal (i.e., day 25).

* Early bleeding started before Day 24 and continued in the expected bleeding period. Continued bleeding started in the expected bleeding period and continued thereafter. DRSP, drospirenone; DNG, dienogest; EE, ethinylestradiol; E2: 17β-estradiol; E2V, E2 valerate; E4, estetrol; FDA, Food and Drug Administration; NETA, norethindrone acetate; NOMAC, nomegestrol acetate.

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
