# Peer review of "Bleeding Patterns of Oral Contraceptives with a Cyclic Dosing Regimen: An Overview"

_jcm, 2022, doi:10.3390/jcm11154634_

Round 1

Reviewer 1 Report

The authors did not use the JCM Microsoft Word template

The authors compared data from phase 3 clinical trials to assess the bleeding risk of oral contraceptives. The data are well presented, in addition, the authors emphasize the importance of using unitary definitions for bleeding, spotting, and scheduled and unscheduled bleeding to be able to objectively compare the risk of OCs available on the market.

The manuscript is well written, but there are some issues that need to be reviewed.

Line 24 – DRSP – is used for the first time in the manuscript, explain the abbreviation 

Line 25 – POP – is used for the first time in the manuscript, explain the abbreviation

Line 82 – please revise the Tables number (Table 2 and 3)

Nice work!

Author Response

Please see also the cover letter.

We greatly appreciate the reviewers’ feedback and your additional thoughts in regard to the manuscript and believe that it has substantially improved the quality of our manuscript.

Comment 1

The authors did not use the JCM Microsoft Word template

Response: We have reformatted the manuscript using the JCM Microsoft Word Template.

Please note that due to reformatting and the use of track changes the line numbers are different in the reformatted manuscript. In our response we refer to the line numbers in the reformatted manuscript.   

Comment 2

The manuscript is well written, but there are some issues that need to be reviewed.

  • Line 24 – DRSP – is used for the first time in the manuscript, explain the abbreviation
  • Line 25 – POP – is used for the first time in the manuscript, explain the abbreviation
  • Line 82 – please revise the Tables number (Table 2 and 3)

Response:

Abbreviations DRSP (drospirenone) and POP (progestin-only pill) have been explained in the abstract. We also checked the use of other abbreviations in the manuscript.

Because some text has been added in the abstract, the term POP is now used for the first time in line 16.

The amended text reads as follows:

  • Lines 13-15: “In this review we provide an overview of bleeding data of recently marketed cyclic combined OCs (COCs) and one progestin-only pill (POP)”.
  • Lines 21-26: “Replacement of EE with estetrol (E4) 15 mg in the combination with drospirenone (DRSP) 3 mg is associated with a predictable and regular scheduled bleeding profile, while the POP containing DRSP 4 mg in a 24/4 regimen is associated with a higher rate of unscheduled and absence of scheduled bleeding than combined products”.

The table numbers in line 86 have been corrected. The text in now reads as follows:

Lines 84-85: “The phase 3 trials with E2/NOMAC included the internal comparator drospirenone 3 mg/EE 30 µg (EE/DRSP), in a 21/7 treatment regimen (Tables 2 and 3).”

In addition to the changes above we made several minor editorial changes, all of which have been tracked.  

Reviewer 2 Report

The review article by David F. Archer focuses evaluated the oral contraceptive-mediated variations in the bleeding pattern. The authors searched PubMed for phase 3 clinical trial-associated publications to assess this. They found that OC comes with different bleeding profiles and, due to the variability in the definitions used for the bleeding patterns, makes comparing products impossible.

The abstract can be improvised; the connecting link is missing while the authors discuss the (line 18) comparison of products and then jump to combined OCs.

“Line 35” The sentence is not clear and can be simplified. “It is often reported at the start of contraceptive use, especially by 36 first-time users, and diminishes, or disappears, after the first three months.”

The authors have used PubMed to search the clinical trials; They can also include the   https://clinicaltrials.gov/ website to search for clinical trials.

Also, they can provide more details on how many hits they got with the above-mentioned search terms, what criteria they chose to filter the search terms, and how they narrowed it down to 13 publications.

In table 2, some places have reported the BMI to be 35, and for some studies, 17 or 18-35. For consistency, they could use one form of representation.  What does Nr in Nr subjects stand for? If it is a number of subjects, they can use No instead of Nr.

Figure:1 X and Y axis are not labeled. On line 152, the authors describe no participants as 76.5; if it is the percent population, then it should be mentioned. The same goes for line 153, 79.7 participants.

The authors have discussed that there is variability in the definition used for the bleeding pattern; they could discuss if anything other than not having clear guidelines could have affected these definitions. 

Author Response

Please see also the cover letter.

We greatly appreciate the reviewers’ feedback and your additional thoughts in regard to the manuscript and believe that it has substantially improved the quality of our manuscript.

Please note that due to reformatting and the use of track changes the line numbers are different in the reformatted manuscript. In our response we refer to the line numbers in the reformatted manuscript.   

Comment 1

The abstract can be improvised; the connecting link is missing while the authors discuss the (line 18) comparison of products and then jump to combined OCs.

Response: We added text on the POP included in this review and specified each type of OC.

  • Old text lines 13-16:

“In this review we provide an overview of bleeding data of recently marketed cyclic OCs. We evaluated data from phase 3 trials (≥12 months) used to gain regulatory approval. Overall, each OC has its own specific bleeding pattern.”

  • Amended text lines 13-16:

“In this review we provide an overview of bleeding data of recently marketed cyclic combined OCs (COCs) and one progestin-only pill (POP). We evaluated data from phase 3 trials (≥12 months) used to gain regulatory approval. Overall, each type of OC has its own specific bleeding pattern.”

Comment 2

“Line 35” The sentence is not clear and can be simplified. “It is often reported at the start of contraceptive use, especially by 36 first-time users, and diminishes, or disappears, after the first three months.”

Response: we have simplified the text in line 35.

  • Old text lines 34-36:

“It is often reported at the start of contraceptive use, especially by first-time users, and diminishes, or disappears, after the first three months [1,2].”

  • Amended text lines 34-36:

“Such bleeding irregularities are often reported at the start of contraceptive use, especially by first time users, and diminish, or disappear after the first three months [1, 2].”

Comment 3:

The authors have used PubMed to search the clinical trials; They can also include the https://clinicaltrials.gov/ website to search for clinical trials.

Also, they can provide more details on how many hits they got with the above-mentioned search terms, what criteria they chose to filter the search terms, and how they narrowed it down to 13 publications.

Response

We primarily based our overview on bleeding data provided in the public assessment reports of regulatory agencies. In addition, we also did a PubMed search to specifically search for publications on the bleeding data of the phase 3 clinical trials that were used for regulatory approval of the 5 oral contraceptive products of interest. During our PubMed search, we discovered that keywords “phase 3” and “bleeding” and “bleeding pattern” were not always used in publications of these phase 3 trials. We therefore did not perform a systematic literature search but performed targeted searches in PubMed to find any peer-reviewed clinical articles that were related to the phase 3 contraceptives trials used to gain regulatory approval. We amended the text in the method section to clarify this. Also, we added text in the discussion section on the use of keywords.      

Clinicaltrials.gov is mentioned and referenced in line 110.

  • Old text lines 89-95:

For the overview we searched for bleeding data from phase 3 contraceptive trials (with a duration of at least 12 months) that were used to gain regulatory approval. From public assessment reports of regulatory agencies, we gathered data on the incidence of scheduled and unscheduled bleeding per treatment cycle and on the number of bleeding days by reference period. Assessment reports sometimes contain more complete and correct data, as publications can come out before approval of product and final corrections in the data are not incorporated.

  • Amended text lines 89-95

For the overview we collected bleeding data from phase 3 contraceptive trials (with a duration of at least 12 months) that were used to gain regulatory approval. From public assessment reports of regulatory agencies, we gathered data on the incidence of scheduled and unscheduled bleeding per treatment cycle and on the number of bleeding days by reference period. Public assessment reports may contain more complete and correct data, as publications can come out before approval of the product in which final corrections are not yet incorporated.

  • Old text lines 103-115

The following search terms were used: oral contraception, oral contraceptive, bleeding, phase 3, drospirenone, nomegestrol, dienogest, norethindrone acetate, ethinylestradiol, estradiol, estradiol valerate, and estetrol. We did not perform a systematic search, but emphasis was placed on articles that could add useful clinical information regarding to the bleeding profile in the phase 3 contraceptive trials. In addition any literature references mentioned in clinical trial.gov register related to these phase 3 clinical trials were assessed. A total of 13 publications with detailed information in the phase 3 studies were included in this overview. Table 2 provides an overview of the trials and sources (public assessment reports and literature) used in this review.

  • Amended text lines 103-115

The following search terms were used: oral contraception, oral contraceptive, bleeding, bleeding pattern, phase 3, drospirenone, nomegestrol, dienogest, norethindrone acetate, ethinylestradiol, estradiol, estradiol valerate, and estetrol. As search terms such as “phase 3” and “bleeding” or “bleeding pattern” were not always used as keywords for the phase 3 clinical trials that were used for registration, we did not perform a systematic search, but emphasis was placed on articles that could add useful clinical information regarding the bleeding profile in these phase 3 contraceptive trials. In addition any literature references mentioned in clinicaltrials.gov register [15] related to these phase 3 clinical trials were assessed. We found 13 publications on the phase 3 clinical trials that provided detailed information on the bleeding profile and included those in this overview. Table 2 provides an overview of the trials and sources (public assessment reports and scientific publications) used in this review.

  • Added text in the discussion, lines 253-258

In our literature search we found that key words for bleeding, bleeding pattern and phase 3 clinical trials were not uniformly applied, making it difficult to perform a systematic review. Instead we chose to perform targeted searches for literature on bleeding reported during the phase 3 clinical trials of the five oral contraceptive products discussed in this review.

Comment 4

In table 2, some places have reported the BMI to be ≤35, and for some studies, 17 or 18-35. For consistency, they could use one form of representation.  What does Nr in Nr subjects stand for? If it is a number of subjects, they can use No instead of Nr.

Response: the details on age and BMI were taken from the details provided by the investigators in their publications, we like to keep this consistent with the publications. A footnote has been added to explain this apparent inconsistency. Nr stands for the number of subject and has been replaced by No. The abbreviation No. has been added as footnote.

Comment 5

Figure:1 X and Y axis are not labeled.

Response: axis titles have been added to the Figure 1: x-axis = Cycle (number of participants); Y-axis = percentage of participants

Comment 6

On line 152, the authors describe no participants as 76.5; if it is the percent population, then it should be mentioned. The same goes for line 153, 79.7 participants.

Response: Percent symbol “%” has been added in lines 160 and 161 as well as at other places where applicable.

Comment 7

The authors have discussed that there is variability in the definition used for the bleeding pattern; they could discuss if anything other than not having clear guidelines could have affected these definitions. 

Response: lines 162 to 169 of the of the result section 3.1 E2V/DNG is actually a discussion on the definitions for bleeding. We therefore decided to move these lines from result section to the discussion section.

  • Deleted text in Section 3.1 - lines 162 tot 169

This high incidence is probably ameliorated by the chosen definition of scheduled bleeding. For E2V/DNG, a scheduled bleeding episode was defined as the first bleeding that started not earlier than Day 21 (i.e., not more than 4 days before DNG withdrawal) and continued without interruption. No end day of the bleeding period was defined. If no bleeding occurred until Day 20 of the next cycle, it was assessed as absence of scheduled bleeding in the previous treatment cycle. Otherwise, any bleeding occurring during the first 20 day of the next cycle caused the previous cycle to be considered as a cycle with scheduled bleeding.

  • Amended text in Discussion section - lines 280 to 290

In phase 3 trials, the use of E2V/DNG resulted in a more stable bleeding pattern, with a relatively high incidence of scheduled bleeding. This high incidence is probably ameliorated by the chosen definition of scheduled bleeding. For E2V/DNG, a scheduled bleeding episode was indeed defined as the first bleeding that started not earlier than Day 21 (i.e., not more than 4 days before DNG withdrawal) and continued without interruption. No end day of the bleeding period was defined. If no bleeding occurred until Day 20 of the next cycle, it was assessed as absence of scheduled bleeding in the previous treatment cycle. Otherwise, any bleeding occurring during the first 20 day of the next cycle caused the previous cycle to be considered as a cycle with scheduled bleeding. The rather unique definition of scheduled bleeding in studies with E2V/DNG evidently increases the proportion of scheduled bleeding episodes and makes a comparison impossible with other OCs using Mishell’s criteria.

In addition to the changes above we made several minor editorial changes, all of which have been tracked.